# Significance of Prediction Models for Post-Hepatectomy Liver Failure Based on Type IV Collagen 7s Domain in Patients with Hepatocellular Carcinoma

**DOI:** 10.3390/cancers16101938

**Published:** 2024-05-20

**Authors:** Takuma Okada, Hiroji Shinkawa, Satsuki Taniuchi, Masahiko Kinoshita, Kohei Nishio, Go Ohira, Kenjiro Kimura, Shogo Tanaka, Ayumi Shintani, Shoji Kubo, Takeaki Ishizawa

**Affiliations:** 1Department of Hepato-Biliary-Pancreatic Surgery, Osaka Metropolitan University Graduate School of Medicine, Osaka 545-8595, Japan; okataku.nov13@gmail.com (T.O.); pikopiko0128@yahoo.co.jp (M.K.); m1155123@omu.ac.jp (K.N.); m1153@omu.ac.jp (G.O.); kenjiro@omu.ac.jp (K.K.); shogotanaka32@gmail.com (S.T.); kubosho65@yahoo.co.jp (S.K.); take1438@gmail.com (T.I.); 2Department of Medical Statistics, Osaka Metropolitan University Graduate School of Medicine, Osaka 545-8595, Japan; taniuchi.satsuki@omu.ac.jp (S.T.); statacademy@med.osaka-cu.ac.jp (A.S.)

**Keywords:** nomogram, serum concentration of type IV collagen 7s domain, post-hepatectomy liver failure

## Abstract

**Simple Summary:**

Currently, a versatile and useful predictive model for post-hepatectomy liver failure (PHLF) remains elusive. Therefore, this study aimed to develop predictive models for PHLF based on type IV collagen 7s domain (7s collagen) in patients with hepatocellular carcinoma. We retrospectively collected data from 972 patients. PHLF grades B or C were identified in 104 patients (11%): 98 (10%) and 6 (1%) PHLF grades B and C, respectively. The preoperative serum level of 7s collagen was significantly associated with a proportional increase in PHLF risk. A nomogram was developed based on 7s collagen, with a concordance index of 0.768. The inclusion of 7s collagen values in the predictive model increased the prediction accuracy. Our novel nomogram using 7s collagen may be useful for predicting the risk of PHLF.

**Abstract:**

*Background:* Previous studies have attempted to establish predictive models for post-hepatectomy liver failure (PHLF) in patients with hepatocellular carcinoma (HCC) undergoing liver resection. However, a versatile and useful predictive model for PHLF remains to be developed. Therefore, we aimed to develop predictive models for PHLF based on type IV collagen 7s domain (7s collagen) in patients with HCC. *Methods:* We retrospectively collected data from 972 patients with HCC who had undergone initial curative liver resection between February 2000 and December 2020 at our hospital. Multivariate logistic regression analysis using a restricted cubic spline was performed to evaluate the effect of 7s collagen on the incidence of PHLF. A nomogram was developed based on 7s collagen. *Results:* PHLF grades B or C were identified in 104 patients (11%): 98 (10%) and 6 (1%) PHLF grades B and C, respectively. Multivariate logistic regression analysis revealed that the preoperative serum level of 7s collagen was significantly associated with a proportional increase in the risk of PHLF, which was confirmed in both laparoscopic and open liver resections. A nomogram was developed based on 7s collagen, with a concordance index of 0.768. The inclusion of 7s collagen values in the predictive model increased the predictive accuracy. *Conclusion:* The findings highlight the efficacy of the serum level of 7s collagen as a predictive factor for PHLF. Our novel nomogram using 7s collagen may be useful for predicting the risk of PHLF.

## 1. Introduction

Hepatocellular carcinoma (HCC) is the sixth most common malignancy and the fourth leading cause of cancer-related death worldwide [1]. Liver resection is a curative treatment with an expected 5-year survival rate of ≥70% [2]. Despite the declining mortality rates of liver resection over the last few decades with improvements in surgical techniques and perioperative management, morbidity rates remain unsatisfactory. Post-hepatectomy liver failure (PHLF) is a severe and fatal complication of liver resection [3,4]. Precise assessment of the risk of PHLF is an important clinical issue. Several risk factors for PHLF have been reported, such as the extent of liver resection, Child–Pugh class, Fibrosis-4 (FIB-4) index, albumin–bilirubin (ALBI) score, indocyanine green (ICG) retention rate at 15 min (ICG15R), advanced fibrosis staging of the background liver, and increased hepatitis activity [5,6,7,8]. Previous studies have attempted to establish predictive models for PHLF in patients with HCC who have undergone liver resection [9,10,11]. However, a versatile and useful predictive model for PHLF remains elusive.

Type IV collagen 7s domain (7s collagen), which is involved in the metabolism of connective tissue, serves as a biochemical marker for fibrogenesis and fibrosis of the liver [12,13,14]. Previously, we reported a correlation between preoperative 7s collagen and PHLF following resection of HCC [15]. Thereafter, several reports indicated that 7s collagen is associated with PHLF, liver regeneration, and liver function recovery [16]. However, the precise contribution of serum 7s collagen levels to the risk of PHLF has not been clarified. Laparoscopic liver resection (LLR), a minimally invasive procedure with a low morbidity rate, has rapidly gained popularity over the last decade [17,18,19]. The effect of differences between LLR and open liver resection (OLR) on the risk of PHLF has not yet been reported.

Therefore, this study aimed to examine the effect of elevated 7s collagen levels on the risk of developing PHLF. Furthermore, we developed a nomogram to predict PHLF, facilitating the quantification and numerical probability of a clinical event, to evaluate the impact of the risk of PHLF on predictive factors.

## 2. Methods

### 2.1. Patients

We retrospectively collected data from 972 patients with HCC who had undergone initial curative liver resection between February 2000 and December 2020 at our hospital. Patients scheduled for biliary reconstruction and/or resection of part of the gastrointestinal tract were excluded. Physical evaluation and medical history interviews were carried out, physical findings were collected, and blood tests were performed 1 or 2 days before surgery (Table 1). This study was approved by the Ethics Committee of our institution (No. 3815) and conducted in accordance with the guidelines of the Declaration of Helsinki.

### 2.2. Measurement of Blood Parameters

Routine laboratory blood tests included measurement of aspartate aminotransferase (AST), alanine aminotransferase (ALT), albumin, and total bilirubin levels in appropriate samples, and usual upper normal values were 30 IU/L, 23 IU/L, 5.1 g/dL, and 1.5 mg/dL, respectively.

The serum 7s collagen level was measured preoperatively using a commercially available type IV collagen 7s domain radioimmunoassay kit (Diaiatron Co., Tokyo, Japan), which uses a polyclonal antibody against the 7s domain of type IV collagen isolated from the human placenta. The reference range of the serum 7s collagen level is ≤6 ng/mL [15].

ICG (Diagnogreen; Daiichisankyo Pharmaceutical Co., Tokyo, Japan; 0.5 mg/kg of body weight) was administered via a peripheral vein, and venous blood was sampled before and 15 min after injection. ICG concentrations of the specimens were analyzed using a spectrophotometer at 805 nm [20].

The FIB-4 index was calculated as follows: age [years] × AST [U/L]/(platelet count [10^9^/L] × ALT[U/L]^1/2^) [21]. The ALBI score was calculated using the following formula: 0.66 × log_10_(bilirubin, μmol/L) − 0.085(albumin, g/L) [22].

### 2.3. Operative Procedures

Liver resection was performed according to Makuuchi’s criteria: presence/absence of ascites, total serum bilirubin level, and ICG15R results [23].

The operative procedures for LLR and OLR were reported previously [24,25]. For OLR, hepatic parenchymal transection was performed using the CUSA Excel Ultrasonic Tissue Ablation System (Integra Life Sciences, Plainsboro, NJ, USA) and an ultrasonically activated device. Hemorrhage from the small intrahepatic vessels was managed via ligation with threads or coagulation using a VIO soft-coagulation system (VIO 300D; ERBE Elektromedizin, Tübingen, Germany). The Pringle maneuver was performed to decrease intraoperative blood loss during liver transection. For LLR, 5–7 trocars were inserted, and the port location was based on the surgery type. Hepatic parenchymal transection was performed using a laparoscopic ultrasonic surgical aspirator, an ultrasonically activated device, and bipolar or monopolar forceps with a VIO soft coagulation system. The Pringle maneuver was performed to decrease intraoperative blood loss during liver transection. The terminology for hepatic anatomy and the type of hepatic resection was based on The Brisbane 2000 Terminology for Liver Anatomy and Resection [26].

### 2.4. Postoperative Management

Patients were allowed to drink on postoperative day (POD) 1 and consume food orally on POD 2. Abdominal drains were removed on POD 2 or 3 after hemostasis, and the absence of bile stains was confirmed.

### 2.5. Definitions of PHLF

PHLF was diagnosed based on the criteria established by the International Study Group of Liver Surgery (ISGLS): total serum bilirubin level of >50 µmol/L and a prothrombin time index of 50% (corresponding to an international normalized ratio (INR) greater than 1.7) on POD 5 or later [3]. Grade A disease was defined as postoperative deterioration of liver function that did not require a change in clinical management [3]. Grade B disease was defined as deviation from the regular, postoperative clinical pathway, but can be managed with noninvasive treatment, including blood transfusion, daily diuretics, and noninvasive ventilation [3]. Grade C disease was defined as PHLF requiring an invasive procedure, including hemodialysis, invasive ventilation, and transplantation [3]. In this study, PHLF was defined as grade B or C according to the ISGLS criteria.

### 2.6. Statistical Analysis

Multivariate logistic regression analysis was performed to evaluate the effect of 7s collagen level on the risk of PHLF, considering the nonlinear association between 7s collagen and PHLF using a restricted cubic spline. The effect was estimated after adjusting for the following covariables: sex, body mass index, ALT, FIB-4 index, ALBI score, Child–Pugh class (A or B), ICG15R, 7s collagen level, surgical approach (LLR or OLR), extent of liver resection, and image-diagnosed portal invasion of the tumor [27]. Extent of liver resection was classified into three categories: ≤1, 2, and ≥3 segments. For missing values, the multiple imputation method was used to generate five sets of imputed datasets with all the explanatory and response variables. In addition to assessing the association, a logistic regression analysis was performed to develop a model for predicting PHLF. Models including the above covariates without/with 7s collagen (Model 1 and Model 2, respectively) were tested to determine how well they discriminated. The performance of the models was evaluated using bootstrap sampling, which was executed before imputing missing values and corrected for “optimism”. Bootstrap validation was performed using 10,000 resamples to validate and calibrate the predictive models. Bootstrap bias-corrected C-indices and calibration line slopes were reported as measures of the predictive performance of the models. To calculate the predicted probabilities of PHLF easily, a nomogram based on Model 2 was developed. The accuracy in predicting the probability of developing PHLF of Models 1 and 2 was compared using net reclassification improvement (NRI) and integrated discrimination improvement (IDI). All statistical analyses were performed with a two-sided significance level of 5% using the R software version 4.0.2 (https://www.r-project.org/ (accessed on 1 January 2022)).

## 3. Results

### 3.1. Patient Characteristics and PHLF

The background characteristics of the 972 patients are shown in Table 1. The median 7s collagen level was 5.7 ng/mL (range: 4.4–7.4). The distribution of the extent of liver resection of ≤1, 2, and ≥3 segments was 671 (69%), 173 (18%), and 128 (13%) patients, respectively. PHLF grade B or C was identified in 104 patients (11%): 98 (10%) and 6 (1%) PHLF grades B and C, respectively.

### 3.2. Association between Preoperative 7s Collagen Level and PHLF

The multivariate logistic regression model revealed that 7s collagen level was significantly associated with a proportional increase in the risk of PHLF (*p* = 0.013) (Figure 1). Furthermore, the risk of PHLF proportionally increased with the 7s collagen level in both LLR and OLR (*p* = 0.017, LLR vs. OLR; *p* for 7s collagen < 0.05) (Figure 2).

### 3.3. Nomogram Predicting PHLF

A nomogram was constructed to predict the probability of PHLF. The established nomogram is shown in Figure 3. By summing the points from each factor, locating the total points on the scale, and drawing a straight line down to the endpoint scales, the nomogram indicated the incidence probability of PHLF.

Age, sex, body mass index, ALT, FIB-4 index, ALBI score, Child–Pugh class (A or B), ICG15R, 7s collagen level, surgical approach (LLR or OLR), extent of liver resection (≤1, 2, and ≥3 segments), and image-diagnosed portal invasion of the tumor were used to build a model to predict the probability of PHLF, as shown below:Pr(PHLF) = 1/(1 + exp[−*Xβ*]), 
where*Xβ* =−3.970229 + 0.01269557 [Age] + 0.427006 [Sex = Male] + 0.009835483 [BMI] + 0.01073821 [ALT] + 0.04651155 [FIB4-index] + 0.5091038 [ALBI score] + 0.00845684 [ICG15R] + 1.255425 [Child Pugh class = B] + 1.09425 [Extent of liver resection = 2 segments] + 0.8149493 [Extent of liver resection ≥ 3 segments] + 0.1569096 [Image diagnosed portal invasion = 3−4] − 1.068218 [Approach = LLR] + 0.08037778 [7s collagen] + 0.002973195 ([7s collagen] − 3.8)^3^_+_ − 0.004634686([7s collagen]−5.7)^3^_+_ + 0.001661491([7s collagen] − 9.1)^3^_+_

(x)_+_ = x if x > 0, 0 otherwise.

### 3.4. Validation of the Proposed Nomogram

The calibration curve showed good agreement between the probability of PHLF predicted using our nomogram and the actual observed risk of PHLF (Figure 4). For the proposed nomogram model, the bootstrap optimism-corrected C-index was 0.768 (95% confidence interval (CI) = 0.720 to 0.814), and the optimism-corrected calibration slope was 0.85 (95% CI = 0.802 to 0.892).

The NRI and IDI were 0.248 (95% CI = −0.022 to 0.491) and 0.020 (95% CI = 0.002 to 0.052), respectively.

## 4. Discussion

In this study, the preoperative serum level of 7s collagen was identified as a significant predictor of a proportional increase in PHLF risk using multivariate logistic regression analyses. We also developed a nomogram for PHLF containing predictive factors, including 7s collagen. This nomogram using 7s collagen was shown to have high accuracy in predicting PHLF, with a bootstrap optimism-corrected C-index of 0.768.

In previous studies, preoperative liver function reserve and the extent of liver resection were shown to be predictive factors for PHLF [5,28,29,30,31,32,33,34]. Liver fibrosis has also been associated with PHLF [35,36]. However, preoperative pathological evaluation of fibrosis staging by liver biopsy is invasive. Thus, noninvasive fibrosis markers have been alternatively used to evaluate background liver fibrosis staging and its relationship with PHLF [37]. Of the fibrosis markers, 7s collagen has been reported to be associated with PHLF [15,16]. However, the extent to which elevated 7s collagen levels affect the risk of developing PHLF has not yet been studied. To make precise treatment decisions based on serum 7s collagen level, we conducted a multivariate analysis and developed a nomogram and found that the risk of PHLF increased with an increase in serum 7s collagen levels on multivariate analysis, after adjustment for background clinical factors. The distribution of points in each predictor shown in the nomogram suggested that 7s collagen contributed as much to the prediction of PHLF as ALT level, ALBI score, FIB-4 index, Child–Pugh class, extent of liver resection, and surgical approach. The scale on the nomogram showed that the point corresponding to a serum 7s collagen value of 11 corresponds to Child–Pugh class B, indicating that these patients were at the same level of risk for PHLF. Furthermore, the inclusion of 7s collagen values in the predictive model increased the prediction accuracy. For patients who are expected to be at high risk for PHLF by this nomogram, a change in treatment strategy, such as a reduction in the extent of liver resection, may be necessary.

Type IV collagen is a basement membrane protein located around sinusoids in the liver [38]. In the normal liver, the basement membrane is not identified in the sinusoids. As liver fibrosis progresses, sinusoidal capillarization occurs, the basement membrane appears, and the serum level of 7s collagen increases [12,13,14]. Several fibrosis markers have been reported to be associated with liver function and PHLF risk, including FIB-4 index, autotaxin [39], and Mac-2 binding protein glycosylation isomer [40]. In this study, fibrosis markers other than 7s collagen and FIB-4 index were not tested and their relationship with PHLF was not examined. However, 7s collagen is involved in the metabolism of connective tissue [12,13,14] and is one of the most sensitive fibrosis markers [12,13,14,16]. A previous report indicated that type IV collagen deposition in the liver rapidly increased from the early to late fibrosis stages [41]. Ishii et al. reported that 7s collagen would reflect fibrogenesis rather than accumulated fibrotic change, which is an indicator of collagen production [16]. Therefore, 7s collagen may be useful in evaluating the risk of PHLF.

LLR, a minimally invasive surgery, has been reported to be associated with a lower incidence of postoperative complications than OLR [42]. In the present study, the preoperative serum level of 7s collagen was significantly associated with a proportional increase in the risk of PHLF in both the LLR and OLR groups. Even in patients with LLR, PHLF should be noted in cases with high serum levels of 7s collagen.

Although ICG15R is a major predictive factor for PHLF, ICG15R points showed a low distribution in the nomogram. A possible reason for this discrepancy is that the hepatic resection type was selected according to Makuuchi’s criteria including the results of the indocyanine green retention test. The extent of hepatic resection (1 and ≥2 sections) had a larger risk scale point in the current nomogram; thus, the predictive significance of ICG15R for PHLF may be diminished by the adopted liver resection type. Overseas liver resection is performed according to the Barcelona Clinic Liver Cancer Staging System, which is different in Japan where portal hypertension is involved in determining the indication for surgery [43]. Consequently, it may be necessary to incorporate serum platelet levels and splenomegaly, factors integral to the analysis, in such investigation.

This study has some limitations. First, the retrospective nature of the study may have resulted in some bias in patient enrollment. Second, we included a relatively small study sample of 972 patients. Third, a resection with ≥3 segments had a smaller risk scale point than with two segments in the current nomogram, although the difference in points was small. Accumulation of more cases and validation of the current nomogram in other facilities worldwide are needed. Finally, our results may have been affected by the long study period. Nevertheless, the predictive value of 7s collagen for PHLF was confirmed even in patients who underwent LLR between January 2008 and December 2020. Furthermore, the long study period at a single institution might have contributed to the evaluation of the significance of 7s collagen based on detailed clinical data.

## 5. Conclusions

This study demonstrated the utility of 7s collagen as a useful predictive factor for PHLF. Treatment strategies for patients with HCC should be based on the expected risk of PHLF. The current nomogram using 7s collagen may be useful for predicting the risk of PHLF.

## Figures and Tables

**Figure 1 cancers-16-01938-f001:**
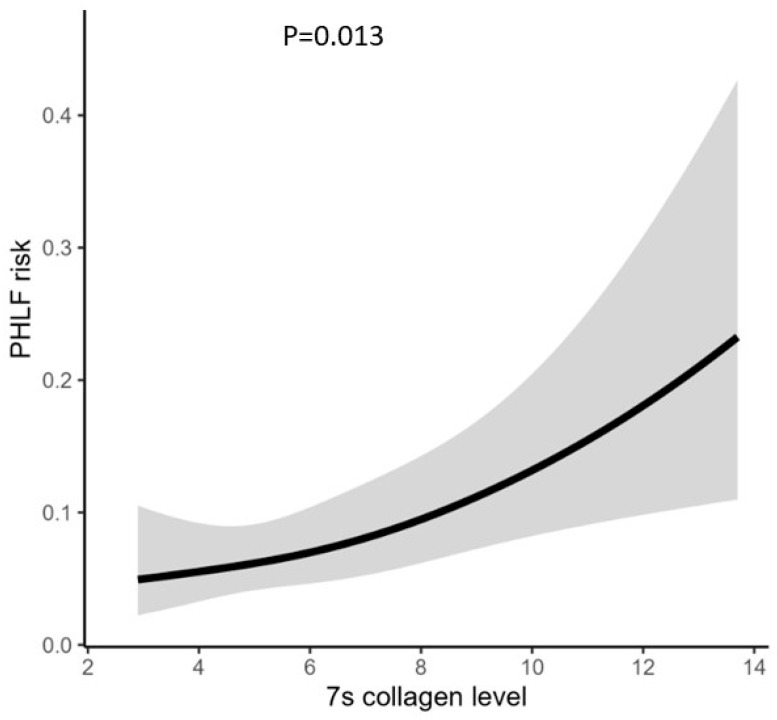
The correlation between 7s collagen level and PHLF risk; PHLF, post-hepatectomy liver failure.

**Figure 2 cancers-16-01938-f002:**
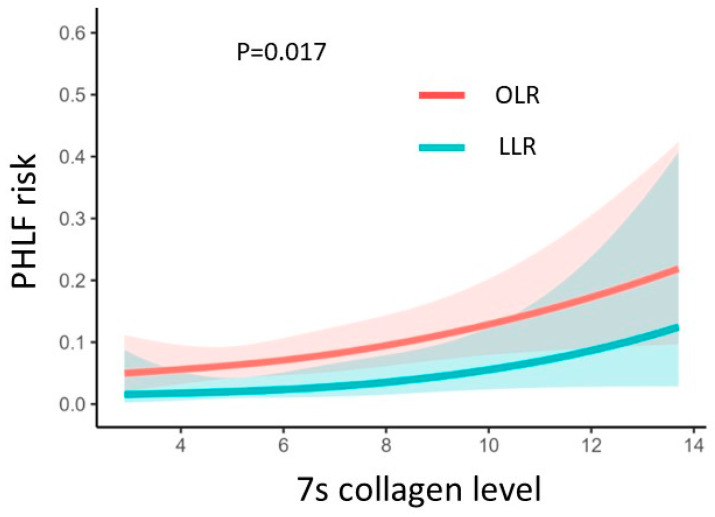
The correlation between 7s collagen level and PHLF risk in LLR and OLR; PHLF, post-hepatectomy liver failure; LLR, laparoscopic liver resection; OLR, open liver resection.

**Figure 3 cancers-16-01938-f003:**
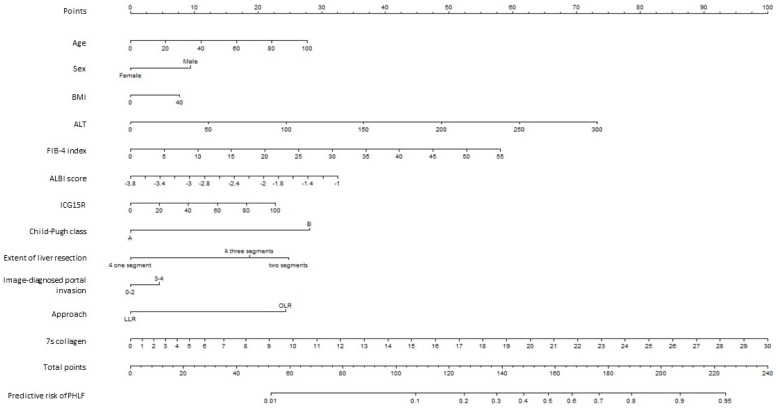
The nomogram to predict the probability of PHLF; PHLF, post-hepatectomy liver failure; BMI, body mass index; ALT, alanine aminotransferase; FIB-4, Fibrosis-4; ALBI, albumin–bilirubin; ICG15R, indocyanine green retention rate at 15 min; LLR, laparoscopic liver resection; OLR, open liver resection.

**Figure 4 cancers-16-01938-f004:**
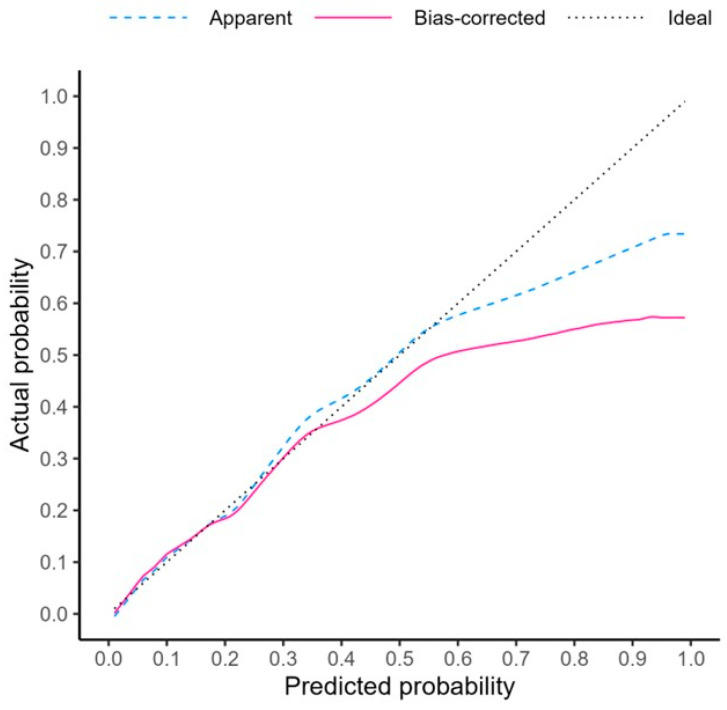
Calibration curves for the predicted probability of PHLF using our nomogram; PHLF, post-hepatectomy liver failure.

**Table 1 cancers-16-01938-t001:** Patient characteristics.

Variables	Patients (*n* = 972)
Sex (male/female)	742/230
Age (y) ^※^	70 (64–75)
BMI (kg/m^2^) ^※^	23.2 (20.9–25.6)
ICG15R (%) ^※^	14.2 (9.8–21.0)
Alanine aminotransferase (IU/L) ^※^	32 (21–53)
Total bilirubin (mg/dL) ^※^	0.7 (0.5–0.9)
Albumin (g/dL) ^※^	4.0 (3.7–4.2)
Prothrombin activity (%) ^※^	93.0 (84.0–102.0)
Child–Pugh class A/B	935/37
FIB-4 index ^※^	2.994 (1.809–4.988)
ALBI score ^※^	−2.685 ((−2.934)–(−2.416))
7s collagen (ng/mL) ^※^	5.7 (4.4–7.4)
Image-diagnosed portal invasion (0–2/3–4)	949/23
Approach (OLR/LLR)	642/330
Extent of liver resection	
≤One segment	671 (69%)
Two segments	173 (18%)
≥Three segments	128 (13%)

^※^ median with interquartile range; BMI, body mass index; ICG15R, indocyanine green retention rate at 15 min.

## Data Availability

The data can be shared up on request.

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
