# Peer review of "Significance of Prediction Models for Post-Hepatectomy Liver Failure Based on Type IV Collagen 7s Domain in Patients with Hepatocellular Carcinoma"

_cancers, 2024, doi:10.3390/cancers16101938_

Round 1
Reviewer 1 Report
Comments and Suggestions for Authors
However, the following issues need to be addressed,
1. The introduction part of the need for the current discussed the research advances in liver cancer surgery, if can consider to some reference documents, such as: https://doi.org/10.1080/21655979.2021.2006552
2. It is recommended to further increase the authentication queue through the public database.
3. The method part needs to further explain the practicability and value of some research methods.
4. The language needs to be polished by professionals
Comments on the Quality of English LanguagePolished MUST BE.
Author Response
Response to Reviewer #1
Thank you very much for your considerate comments.
We have revised the manuscript according to your comments.
Comment 1. The introduction part of the need for the current discussed the research advances in liver cancer surgery, if can consider to some reference documents, such as: https://doi.org/10.1080/21655979.2021.2006552
Response: Thank you for the comment. We have incorporated information about recent advancements in liver cancer surgery, supported by citations from recent literature, including the suggested document in the Introduction section. (Page 5, lines 3–12)
Comment 2. It is recommended to further increase the authentication queue through the public database.
Response: Thank you for the comment. We have included the database for the current study as a supplementary table.
Comment 3. The method part needs to further explain the practicability and value of some research methods.
Response: Thank you for the comment. We have incorporated a method for the preoperative assessment and measurement of blood parameters, including biochemistry tests, 7s collagen, and indocyanine green (ICG) tests. (Page 6, lines 17–19, page 7 lines 5–16, and page 10 lines 4–7)
Comment 4. The language needs to be polished by professionals
Response: We enlisted the help of a native English speaker to proofread our text.

Reviewer 2 Report
Comments and Suggestions for Authors
In this manuscript, the authors developed prediction models for post-hepatectomy liver failure (PHLF) based on type â…£ collagen 7s domain (7s collagen) in patients with hepatocellular carcinoma (HCC). 972 HCC patients who underwent curative liver resection were recruited. The authors found that PHLF grades B or C were identified in 104 patients (11%), the preoperative serum level of 7s collagen was significantly associated with a proportional increase in PHLF risk in both laparoscopic and open liver resections, alanine aminotransferase, Child–Pugh class B, 7s collagen, open liver resection, and extent of liver resection were found to be independent risk factors for PHLF by multivariate logistic regression analysis, a nomogram based on 7s collagen was found with a concordance index of 0.762, and that the inclusion of 7s collagen values in the prediction model significantly increased the prediction accuracy. So the authors concluded that the current nomogram using 7s 30 collagen may be useful for predicting the risk of PHLF.
This is a retrospective study to develop prediction models for post-hepatectomy liver failure based on type â…£ collagen 7s domain in HCC patients. The data collection was not so complete. The data analysis and interpretation were not appropriate. However, this article can provide useful information for the clinicians to manage HCC patients with post-hepatectomy.
Comments
1. Regarding the risk factors for post-hepatectomy liver failure, the functional reserve and severity of liver fibrosis are very important. Besides ICG15R, total bilirubin, albumin, prothrombin activity, Child–Pugh class A/B and 7s collagen, the authors should evaluate the roles of ALBI grade (for functional reserve) and non-invasive measurement of liver fibrosis (e.g. Fibroscan, FIB-4) in the development of post-hepatectomy liver failure, and compare the prediction accuracies between the 7s collagen-based model and non-invasive measurement of liver fibrosis-based model.
2. The authors should describe the methods for 7s collagen detection, serum biochemistry, and ICG15R examination.
3. The authors should cite more recent references.
4. The English needs polishing.
Comments on the Quality of English LanguageThe English needs polishing.
Author Response
Response to Reviewer #2
Thank you very much for your considerate comments.
We have revised the manuscript according to your comments.
Comment 1. Regarding the risk factors for post-hepatectomy liver failure, the functional reserve and severity of liver fibrosis are very important. Besides ICG15R, total bilirubin, albumin, prothrombin activity, Child–Pugh class A/B and 7s collagen, the authors should evaluate the roles of ALBI grade (for functional reserve) and non-invasive measurement of liver fibrosis (e.g. Fibroscan, FIB-4) in the development of post-hepatectomy liver failure, and compare the prediction accuracies between the 7s collagen-based model and non-invasive measurement of liver fibrosis-based model.
Response: Thank you for the comment. We refined our predictive models by incorporating the fibrosis-4 index and albumin-bilirubin score for the multivariate logistic regression model to evaluate the association between preoperative 7s collagen level and post-hepatectomy liver failure (PHLF). Additionally, we developed a nomogram to predict the likelihood of PHLF. A multivariate logistic regression model revealed that 7s collagen level was significantly associated with a proportional increase in PHLF risk (P = 0.013) (Fig 1). The risk of PHLF proportionally increased with the 7s collagen level in both LLR and OLR (P = 0.017, LLR vs. OLR; P for 7s collagen < 0.05) (Fig 2). (Page 11, lines 14–17) Furthermore, the nomogram with the 7s collagen was demonstrated to have high accuracy in predicting PHLF, with a bootstrap optimism-corrected C-index of 0.768. The current nomogram using 7s collagen demonstrated an increased accuracy of NRI 0.248 (95% confidence interval [CI] = -0.022 to 0.491) and IDI 0.020 (95% CI = 0.002–0.052) (Fig 3). (Page 12, lines 1–4)
In the multivariate analysis performed during the creation of the nomogram, because a single odds ratio cannot summarize the effect of the nonlinear term, a predictive formula is presented in the results section. (Page 12, lines 5–18) The results of the multivariate analysis in Table 2 were eliminated. We have revised the discussion accordingly. (Page 13, lines 13 and 15; Page 14, lines 8–11; Page 15 lines 17 and 18; Page 16 lines 1, 2, and 11)
Comment 2. The authors should describe the methods for 7s collagen detection, serum biochemistry, and ICG15R examination.
Response: Thank you for the comment. We have included a method for measuring blood parameters, including biochemistry tests, 7s collagen, and ICG tests. (Page 6, lines 17–19; Page 7, lines 5–16)
Comment 3. The authors should cite more recent references.
Response: Thank you for the comment. We have incorporated recent references outlining advancements in liver cancer surgery. (Page 5, lines 3–12)
Comment 3. The English needs polishing.
Response: We had a native English speaker proofread our text.

Round 2
Reviewer 1 Report
Comments and Suggestions for Authors
Accept.